# Singlet and triplet to doublet energy transfer: improving organic light-emitting diodes with radicals

Feng Li [1,2,6], Alexander J. Gillett [2,6], Qinying Gu[2], Junshuai Ding[1], Zhangwu Chen[1], Timothy J. H. Hele [3], William K. Myers [4], Richard H. Friend [2✉] & Emrys W. Evans [2,5✉]

Organic light-emitting diodes (OLEDs) must be engineered to circumvent the efficiency limit imposed by the 3:1 ratio of triplet to singlet exciton formation following electron-hole capture. Here we show the spin nature of luminescent radicals such as TTM-3PCz allows direct energy harvesting from both singlet and triplet excitons through energy transfer, with subsequent rapid and efficient light emission from the doublet excitons. This is demonstrated with a model Thermally-Activated Delayed Fluorescence (TADF) organic semiconductor, 4CzIPN, where reverse intersystem crossing from triplets is characteristically slow (50% emission by 1 μs). The radical:TADF combination shows much faster emission via the doublet channel (80% emission by 100 ns) than the comparable TADF-only system, and sustains higher electroluminescent efficiency with increasing current density than a radical-only device. By unlocking energy transfer channels between singlet, triplet and doublet excitons, further technology opportunities are enabled for optoelectronics using organic radicals.

[1] State Key Laboratory of Supramolecular Structure and Materials, College of Chemistry, Jilin University, Qianjin Avenue 2699, Changchun 130012, P. R. China. [2] Cavendish Laboratory, University of Cambridge, JJ Thomson Avenue, Cambridge CB3 0HE, UK. [3] Department of Chemistry, University College London, Christopher Ingold Building, London WC1H 0AJ, UK. [4] Centre for Advanced Electron Spin Resonance (CAESR), Department of Chemistry, University of Oxford, Inorganic Chemistry Laboratory, South Parks Road, Oxford OX1 3QR, UK. [5] Department of Chemistry, Swansea University, Singleton Park, Swansea SA2 8PP, UK. [6] These authors contributed equally: Feng Li, Alexander J. Gillett. ✉email: rhf10@cam.ac.uk; emrys.evans@swansea.ac.uk

Spin management is an important consideration for organic light-emitting diode (OLED) efficiency in display and lighting technologies. For closed-shell molecules with singlet-spin-0 ground state, spin statistics with electrical excitation leads to the formation of 25% singlet (spin-0, $S_1$) and 75% triplet (spin-1, $T_1$) excitons[1,2]. In first-generation OLEDs, this results in maximum electroluminescence (EL) internal quantum efficiency (EQE) of 25% as singlet emission (fluorescence, $S_1 \rightarrow S_0 + h\nu$) is allowed whereas triplet emission (phosphorescence, $T_1 \nrightarrow S_0 + h\nu$) is spin-forbidden. In commercial applications, triplet–triplet annihilation- and enhanced phosphorescence-based schemes have been used to obtain efficient luminescence from triplet states[3–7]. Other technologies under development include thermally activated delayed fluorescence (i.e., TADF)[8–12], where electron donor–acceptor molecular designs promote reduced exchange interaction and minimised $S_1$-$T_1$ energy gap for reverse intersystem

crossing (rISC, $T_1 \rightarrow S_1$) and delayed $S_1$ emission. The TADF electroluminescence mechanism is shown in Fig. 1a.

Another possibility to extract emission from the otherwise dark $T_1$ state is to transfer its energy to another energy acceptor molecule, which then emits light. However, if the acceptor is a ground-state singlet, converting the donor triplet to an acceptor excited-state emissive singlet is spin-forbidden:

$$D(T_1) + A(S_0) \nrightarrow D(S_0) + A(S_1) \quad (1)$$

where $D(X)$ stands for the energy donor molecule in state $X$ and $A(X)$ for the energy acceptor, and $\rightarrow/\nrightarrow$ denotes spin-allowed/forbidden. It is possible to convert the donor triplet to an acceptor triplet, but emission from this state is spin-forbidden

$$D(T_1) + A(S_0) \rightarrow D(S_0) + A(T_1) \nrightarrow D(S_0) + A(S_0) + h\nu \quad (2)$$

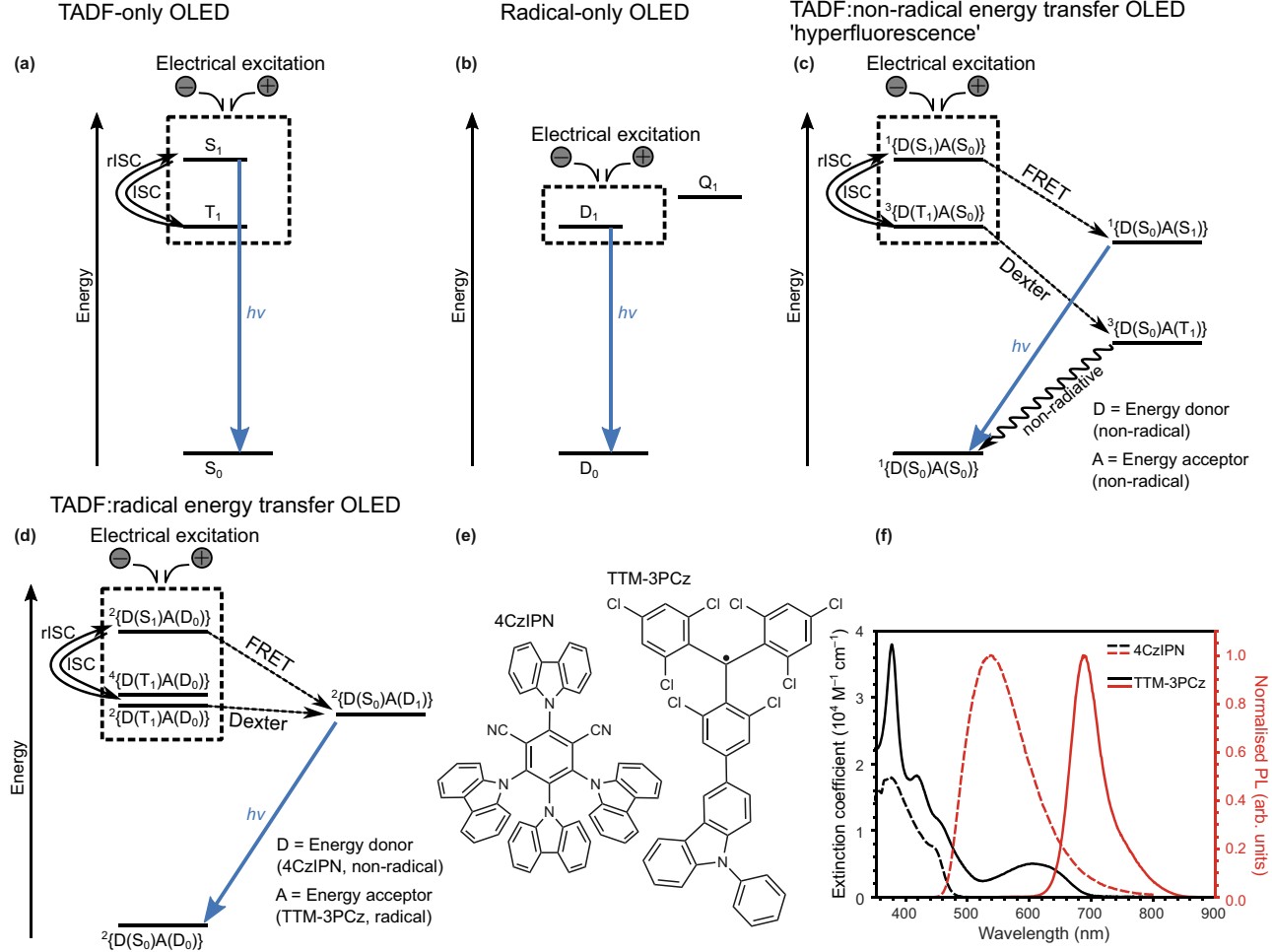

**Fig. 1 Light emission mechanisms and the radical energy transfer system.** Electroluminescence mechanisms for TADF-only, radical-only and energy transfer OLEDs. Spin-allowed radiative transitions from excited to ground states are indicated by blue arrows labelled 'hv.' **a** Scheme for TADF OLED mechanism with emission from singlet $S_1$ exciton, and singlet–triplet intersystem crossing (ISC) and reverse intersystem crossing (rISC) processes with non-emissive triplet $T_1$ exciton. **b** Scheme for radical OLED mechanism with emission from doublet $D_1$ exciton, formed by direct electrical excitation. Higher energy and non-emissive quartet $Q_1$ exciton state are shown. **c** Scheme for TADF:non-radical energy transfer OLED mechanism. Electrical excitation generates singlet $D(S_1)$ and triplet $D(T_1)$ excitons, with FRET singlet-singlet energy transfer to non-radical energy acceptor (A) to form emissive singlet excitons, $A(S_1)$. Dexter triplet–triplet energy transfer forms non-emissive triplet excitons, $A(T_1)$; non-radiative decay to the ground state is shown by a wavy arrow. ISC and rISC steps between $D(T_1)$ and $D(S_1)$ are indicated. Spin multiplicity of D and A pairs are denoted by $2S+1$ in $^{2S+1}\{D\ A\}$. **d** Scheme for TADF:radical energy transfer OLED mechanism. Electrical excitation generates singlet $D(S_1)$ and triplet $D(T_1)$ excitons, with singlet–doublet FRET and triplet–doublet Dexter energy transfer to radical energy acceptor (A) to form emissive doublet excitons, $A(D_1)$. **e** Chemical structures for 4CzIPN and TTM-3PCz used to test the mechanism in (**d**). **f** Absorption (black) and normalised PL (red) profiles for 4CzIPN (dotted lines) and TTM-3PCz (solid lines).

such that the process converts one dark state to another dark state. A donor singlet can transfer its energy to the acceptor singlet by Förster transfer:

$$D(S_1) + A(S_0) \rightarrow D(S_0) + A(S_1) \rightarrow D(S_0) + A(S_0) + h\nu \tag{3}$$

but since this converts one bright state to another bright state, it does not improve the device efficiency, though could improve other device characteristics such as colour purity. Singlet to singlet energy transfer has been achieved, in previous work[13–17], where TADF materials have been used as sensitisers in Förster-type energy transfer of TADF $S_1$ to non-radical fluorescent molecules in a 'hyperfluorescence' scheme as depicted in Fig. 1c. In these systems, energy transfer of TADF triplet excitons is indirect and proceeds following reverse intersystem crossing to the TADF $S_1$. However, the undesirable triplet–triplet energy transfer to lower energy triplets on the 'hyperfluorescent' molecule, as mentioned above, as well as undesirable triplet-annihilation interactions, must therefore be suppressed.

In contrast to OLED technologies employing electronic excitations with paired electrons, efficient radical-based OLEDs offer an alternative route to overcoming the spin-statistics limit using doublet excitons with spin-allowed doublet emission ($D_1 \rightarrow D_0 + h\nu$ fluorescence), since the dark quartet state $Q_1$ lies above the $D_1$ state in energy[18–24] (note that $D_x$ denotes doublet electronic states, and D denotes energy donor). The radical OLED photophysical mechanism is shown in Fig. 1b. However, despite demonstrating an excellent peak EQE at low injection current densities, the 'roll-off'—decreasing efficiency with increasing current density—is severe in reported radical devices using single-dopant emissive layers where charge trapping directly forms doublet excitons[20,25]. The role of exciton-exciton and exciton-charge annihilation effects were ruled out by transient PL measurements on electrically-driven OLEDs, leading to the conclusion that the charge-trapping mechanism for EL must be improved to advance the performance of radical-based devices[25].

Here we consider if the desirable properties of radical emitters could be used to 'brighten' otherwise dark (or slowly emissive) triplet states where emission efficiency cannot easily be improved by using a ground-state singlet acceptor. In the SI section 1, we show how, using a ground-state radical acceptor, triplet energy transfer leading to an emissive excited-state doublet can be quantum mechanically spin-allowed by Dexter transfer:

$$D(T_1) + A(D_0) \rightarrow D(S_0) + A(D_1) \rightarrow D(S_0) + A(D_0) + h\nu \tag{4}$$

unlike the case of a ground-state singlet acceptor considered earlier. Energy transfer from an excited-state singlet to a doublet is also allowed via a Förster-type mechanism

$$D(S_1) + A(D_0) \rightarrow D(S_0) + A(D_1) \rightarrow D(S_0) + A(D_0) + h\nu \tag{5}$$

meaning that the radicals' doublet-spin nature enables energy harvesting of all electronic excitations in standard organic semiconductors. In addition, rapid EL emission can be enabled in radical energy transfer-based devices, which is desirable: to enhance EL efficiency in OLEDs by outcompeting non-radiative channels, and to avoid building up of high excitation densities at high drive currents that can cause efficiency roll-off. Previously, triplet to doublet energy transfer has been demonstrated in experiments using transient radical acceptors[26], but to the best of our knowledge has not been demonstrated using a stable, emissive radical nor in an optoelectronic device.

We have combined non-radical organic semiconductors as energy donors with radical emitters as energy acceptors to form light-emitting layers. In principle, the strategy we propose can work with a wide range of standard OLED semiconductors so long as their singlet and triplet states are higher in energy than the doublet exciton in the radical material. It is desirable to choose systems for which the spin-exchange energy is kept low, so that the singlet energy is kept low, and (as in the case of 'hyperfluorescence' mentioned earlier) we use here TADF materials that are engineered to reduce the exchange energy to thermally accessible values. A further advantage here is that TADF systems undergo intersystem crossing following photo-excitation, allowing us to follow singlet and triplet dynamics in transient all-optical measurements. Thus our energy donors and acceptors in double-dopant emissive layers were chosen to be the benchmark TADF material, 1,2,3,5-tetrakis(carbazole-9-yl)-4,6-dicyanobenzene (4CzIPN)[8], and tris(2,4,6-trichlorophenyl) methyl-3-substituted-9-phenyl-9H-carbazole (TTM-3PCz) radical from our previous work[20]. Transient PL (trPL) and absorption (TA) measurements were used to probe the singlet–doublet and triplet–doublet energy transfer mechanisms, showing rapid energy transfer on picosecond and microsecond timescales from singlet and triplet excitons, respectively. Magneto-electroluminescence studies support the role of triplet–doublet energy transfer in radical-based OLEDs. The TADF:radical devices show improved device characteristics, with reduced turn-on voltage and roll-off in the EQE, as well as better device stability than single-dopant radical structures. TADF:radical systems extend the spin space of organic optoelectronics, where advantageous 'hyperfluorescence' can be retained, dark triplet states removed, and more direct triplet–doublet energy transfer used for efficient radical-based optoelectronics.

## Results and discussion

**Radical energy harvesting for doublet emission**. Figure 1d shows an energy level diagram for radical-based OLEDs using double-dopant emissive layers containing non-radical organic components (D, energy donor) and radical emitters (A, energy acceptor). General design rules are formulated: singlet ($S_1$) and triplet ($T_1$) excitons of D can transfer energy to the doublet ($D_1$) of A for efficient doublet emission where

1. The singlet and triplet energy levels of the donor are higher than the $D_1$ state of the acceptor, i.e., $E(D, S_1) > E(A, D_1)$ and $E(D, T_1) > E(A, D_1)$ where $E(D, S_1)$ and $E(D, T_1)$ are the $S_1$ and $T_1$ exciton energies of D, and $E(A, D_1)$ is the radical A $D_1$ exciton energy;
2. The donor-cation/acceptor-anion, $D^{\bullet+} A^{\bullet-}$ or donor-anion/acceptor-cation, $D^{\bullet-} A^{\bullet+}$ states must be higher energy than the radical $D_1$-exciton, i.e., $E(D^{\bullet+} A^{\bullet-}) > E(A, D_1)$ and $E(D^{\bullet-}A^{\bullet+}) > E(A, D_1)$.

As energy donors and acceptors, 4CzIPN ($E(D,$ HOMO$) = -5.8$ eV; $E(D,$ LUMO$) = -3.4$ eV)[27] and TTM-3PCz ($E(A,$ HOMO$) = -5.8$–$6$ eV; $E(A,$ SOMO reduction$) = -3.7$ eV)[20] were chosen, and their molecular structures are given in Fig. 1e. Singlet–doublet transfer (Fig. 1d, dotted arrow) by a dipolar fluorescence resonance energy transfer, FRET, mechanism results in conservation of doublet-spin multiplicity from $^2S_1$ to $^2S_0$. This was promoted by spectral overlap of TTM-3PCz A-absorption and D-fluorescence of 4CzIPN (Fig. 1f), a well-studied TADF emitter with a singlet–triplet exchange energy gap of <50 meV[28,29]. The small singlet–triplet energy gap also allows substantial spectral overlap of D-phosphorescence and A-absorption, which also leads to a resonant energy condition. This sets up conditions for triplet–doublet energy transfer by electron-exchange Dexter mechanism (Fig. 1d, dotted arrow) from long-lived (>microsecond) 4CzIPN triplet excitons, which can be harvested for light emission.

The reverse process—doublet to triplet energy transfer—was previously demonstrated by us and others with TTM-carbazole and anthracene derivatives[30]. Triplet–doublet energy transfer to form $^2S_0$ is spin-allowed by the $^2T_1$ state, which is mixed with the $^4T_1$ state because of the negligible doublet–quartet $^{2,4}T_1$ energy difference (estimated to be ~10 μeV from the intermolecular approach with no bond formation where antiferromagnetic coupled doublet is the lowest energy state[31], meaning they are effectively degenerate) and spin mixing terms such as the triplet zero-field splitting interaction[32]. The mixed $^{2,4}T_1$ states allow unlocked triplet–doublet channels for direct energy transfer with organic radicals. The theoretical considerations for singlet–doublet and triplet–doublet energy transfer by FRET and Dexter mechanisms are discussed further in Supplementary Information 1.

**Energy transfer photophysics with radical emitters.** In order to understand the photophysics of combined TADF:radical materials we firstly studied films that were radical-only, TADF-only and TADF:radical blends. We used time-resolved optical spectroscopy measurements to probe energy transfer from 4CzIPN to TTM-3PCz on pico- to microsecond timescales. The film composition for studying the radical energy transfer concept was 4CzIPN:TTM-3PCz:CBP (ratio = 0.25:0.03:0.72). Reference films were studied for TTM-3PCz radical only (TTM-3PCz:CBP, 0.03:0.97) and 4CzIPN TADF only (4CzIPN:CBP, 0.25:0.75). The composition is based on the starting point of our previous work on TTM-3PCz OLEDs[20], which here allows us to test energy transfer mechanisms in proof-of-principle studies. 4CzIPN and TTM-3PCz were blended in CBP (4,4'bis(N-carbazolyl)-1,1'-biphenyl) to reduce the effects of exciton self-quenching[33], and with higher doping of 4CzIPN than the radical to promote charge trapping at the TADF sites and subsequent energy transfer to TTM-3PCz for light emission.

TrPL profiles for nano-to-microsecond time ranges (with 355 nm excitation, all fluences = 5 μJ/cm$^2$) of 4CzIPN:TTM-3PCz:CBP films are found to be superpositions of TTM-3PCz (~700 nm) and 4CzIPN (~530 nm) emission. PL timeslices (2.5 ns) are given in Fig. 2a for 4CzIPN:TTM-3PCz:CBP (red), 4CzIPN:CBP (black) and TTM-3PCz:CBP (blue). In Fig. 2b, normalised PL spectra with respect to radical emission (timeslices from 2.5 to 50 ns) show substantial quenching of 4CzIPN on nanosecond timescales. For OLED applications it is desirable to reduce the overall emission time to minimise exciton quenching

mechanisms[34], leading us to consider plots of the integrated PL fraction for total emission (Fig. 2c). From this, we observe in 4CzIPN:TTM-3PCz:CBP that 95% of all photons are emitted by 1 μs, and over 80% of emission occurring by 100 ns. This compares favourably to 4CzIPN:CBP where only ~50% of emission happens by 1 μs, such that the donor–acceptor blend shows faster emission than the 4CzIPN-only blend.

We have performed TA studies of 4CzIPN:TTM-3PCz:CBP, TTM-3PCz:CBP and 4CzIPN:CBP films in order to elucidate the energy transfer processes from excited-state absorption kinetics. In Fig. 3a, $\Delta T/T$ spectral timeslices are presented for short-time TA of 4CzIPN:TTM-3PCz:CBP from 0.2–0.3 ps to 1000–1700 ps. Excitation at 400 nm allowed for the preferential formation of excitons on 4CzIPN, owing to its strong absorption in this region and significantly higher loading fraction. The initial TA spectrum of 4CzIPN:TTM-3PCz:CBP (0.2–0.3 ps) closely resembles that of 4CzIPN:CBP, where we have assigned the 4CzIPN ground-state bleach between 360–460 nm, the 4CzIPN stimulated emission overlaid on a photoinduced absorption (PIA) between 480 and 700 nm, and the primary 4CzIPN $S_1$ PIA at 830 nm (see Supplementary Figs. 1 and 2 for TA of 4CzIPN:CBP films). By 10 ps, we observe new PIA bands that grow in for 4CzIPN:TTM-3PCz:CBP at 620, 950 and 1650 nm. These features match with the TTM-3PCz $D_1$ spectral profile obtained from studies of TTM-3PCz:CBP films (Supplementary Figs. 3 and 4), showing energy transfer from TADF singlet to radical doublet. In Fig. 3b, the normalised $\Delta T/T$ kinetic profiles for 4CzIPN:TTM-3PCz:CBP in TTM-3PCz $D_1$ (610–630 nm, red line) and 4CzIPN $S_1$ (800–830 nm, orange line) PIA regions are shown. We highlight an additional quenching of 4CzIPN in 4CzIPN:TTM-3PCz:CBP compared to 4CzIPN:CBP films (black line, Fig. 3b). The quenching of 4CzIPN $S_1$ PIA and the growth of TTM-3PCz $D_1$ PIA on picosecond timescales prior to nanosecond 4CzIPN intersystem crossing is attributed to Förster-type singlet–doublet energy transfer[35]. As the 4CzIPN $S_1$ PIA lies in a region where there is reduced absorption by the TTM-3PCz $D_1$, we can use the $\Delta T/T$ with and without the presence of TTM-3PCz to estimate a lower bound for the fraction of singlet–doublet energy transfer. By 1.7 ns, the 4CzIPN $S_1$ PIA falls to approximately 45% and 60% of the initial signal with (orange) and without (black) TTM-3PCz present, respectively, suggesting that ≥15% of $S_1$ from 4CzIPN have already undergone fluorescence resonance energy transfer (FRET) to TTM-3PCz in 4CzIPN:TTM-3PCz:CBP. With selective excitation of TTM-3PCz at 600 nm (below the 4CzIPN bandgap)

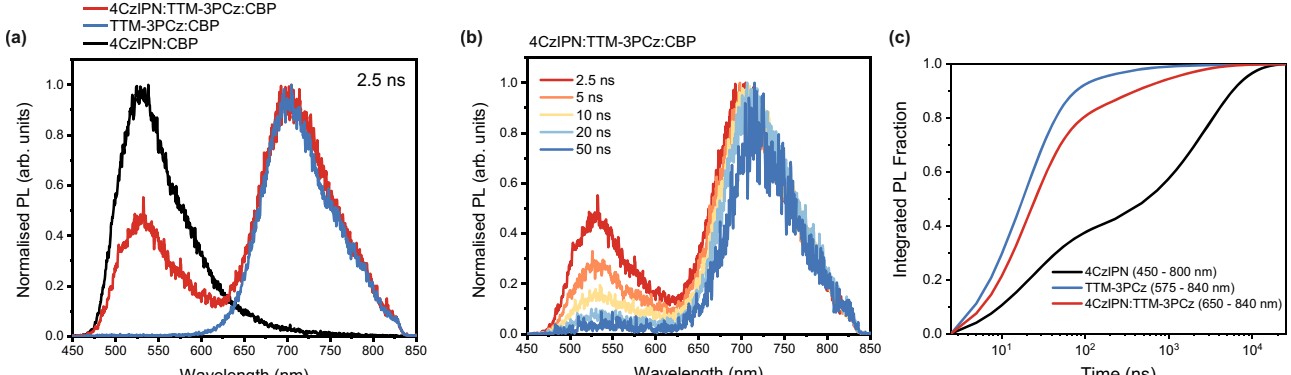

**Fig. 2 Transient photoluminescence studies of 4CzIPN and TTM-3PCz with 355 nm excitation. a** PL timeslices at 2.5 ns for 4CzIPN:TTM-3PCz:CBP (ratio = 0.25:0.03:0.72, red line); 4CzIPN:CBP (0.25:0.75, black line); TTM-3PCz:CBP (0.03:0.97, blue line), showing emission from both TADF and radical in the combined film. **b** PL timeslices for 4CzIPN:TTM-3PCz:CBP at various times from 2.5 to 50 ns, showing the 4CzIPN emission decaying relative to the radical emission at longer times. **c** Integrated PL fraction time profiles from 2.5 ns to 25 μs for 4CzIPN:TTM-3PCz:CBP in 650–840 nm range (red line); 4CzIPN:CBP in 450–800 nm range (black line); and TTM-3PCz:CBP in 575–840 nm range (blue line), showing faster luminescence for the combined TADF:radical film than the TADF-only film.

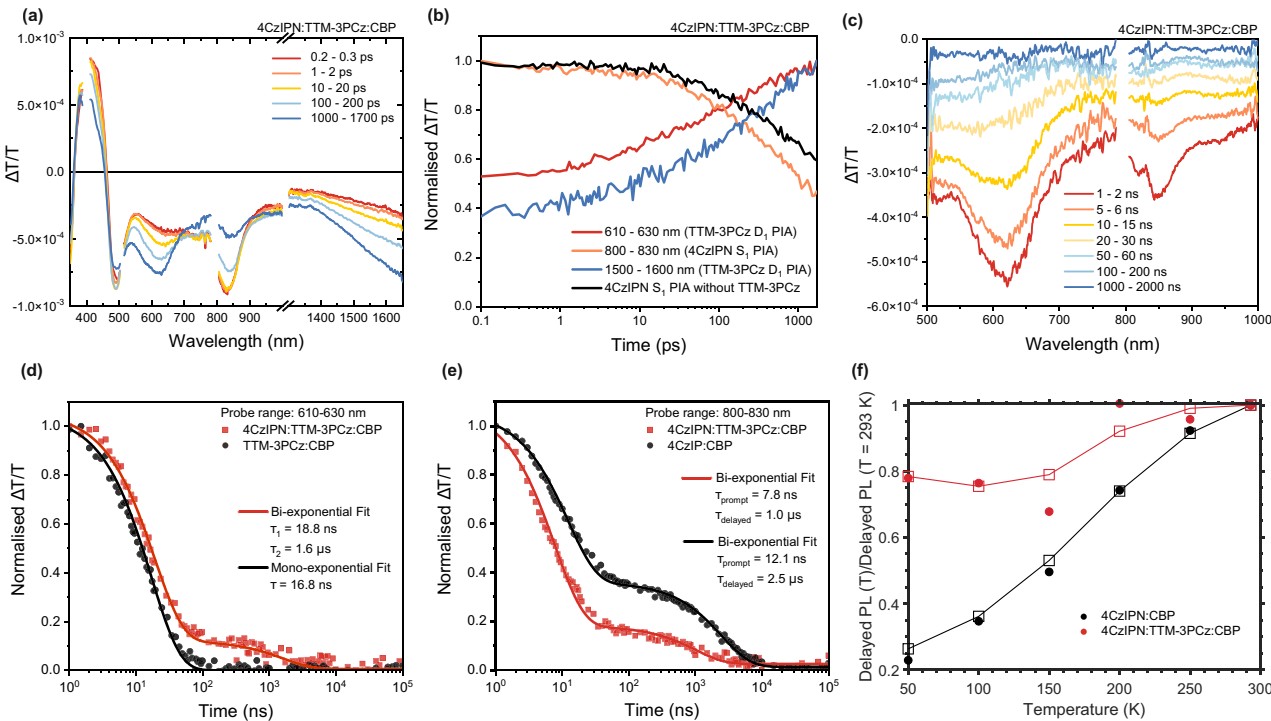

**Fig. 3 Transient absorption and temperature dependence studies of 4CzIPN and TTM-3PCz.** Picosecond to nanosecond (**a**) timeslices and (**b**) kinetic profiles from transient absorption studies of 4CzIPN:TTM-3PCz:CBP (ratio = 0.25:0.03:0.72). 400 nm excitation, fluence = 89.1 μJ/cm². This shows the decay of the singlet PIA around 830 nm and the growth of the radical PIAs around 620 and 1650 nm. **c** Nanosecond to microsecond timeslices of the 4CzIPN:TTM-3PCz:CBP blend (0.25:0.03:0.72). 355 nm excitation, fluence = 17.0 μJ/cm². Discontinuities in timeslice spectral profiles for (**a**) and (**c**) arise because multiple experiments are used to cover the studied wavelength probe regions. Transient absorption kinetic profiles for photoinduced absorption features of (**d**) TTM-3PCz (610–630 nm) and (**e**) 4CzIPN (800–830 nm). **d** TTM-3PCz excited-state kinetics are shown for 4CzIPN:TTM-3PCz:CBP (0.25:0.03:0.72, red squares); and TTM-3PCz:CBP (0.03:0.97, black circles). This shows delayed radical emission is active in 4CzIPN:TTM-3PCz:CBP (TADF:radical) from triplet–doublet energy transfer. **e** 4CzIPN excited-state kinetics are shown for 4CzIPN:TTM-3PCz:CBP (red squares); and 4CzIPN:CBP (0.25:0.75, black circles). This shows delayed radical emission in 4CzIPN:TTM-3PCz:CBP (TADF:radical) is more rapid than delayed emission in 4CzIPN:CBP (TADF only). Mono- and bi-exponential fits are indicated by solid lines in (**d** and **e**). **f** Ratio of integrated delayed PL contribution for 4CzIPN:CBP (black circles) and 4CzIPN:TTM-3PCz:CBP (red circles) at different temperatures. Three-point moving average and trends for these profiles are indicated by square and line plots, and show different temperature dependencies.

in 4CzIPN:TTM-3PCz:CBP, the resulting TA profiles resemble TTM-3PCz:CBP, showing that the $D_1$ exciton—once formed—does not interact with 4CzIPN by further energy or charge transfer processes (Supplementary Figs. 5 and 6).

We have studied energy transfer for timescales beyond 1 ns with long-time TA measurements of 4CzIPN:TTM-3PCz:CBP films (excited at 355 nm). $\Delta T/T$ spectral timeslices (1–2 ns to 1000–2000 ns) in Fig. 3c display features at 620, 830 and 1600 nm, which can be attributed to the TTM-3PCz $D_1$ PIA and 4CzIPN $S_1$ PIA from radical-only (Supplementary Figs. 3 and 4) and TADF-only films (Supplementary Figs. 1 and 2). The kinetic decay profile of the TTM-3PCz PIA (600–630 nm) has an extended lifetime in 4CzIPN:TTM-3PCz:CBP films (red squares, Fig. 3d) compared to TTM-3PCz:CBP (black circles). The 4CzIPN:TTM-3PCz:CBP kinetic profile can be fitted to a bi-exponential with time constants of $\tau_1$ = 18.8 ns and $\tau_2$ = 1.6 μs. The presence of a long-lived $D_1$ state in 4CzIPN:TTM-3PCz:CBP, beyond the $D_1$ excited-state lifetime measured from TTM-3PCz:CBP ($\tau$ = 16.8 ns, Supplementary Fig. 4), suggests energy transfer from 4CzIPN triplet ($T_1$) states. By comparing the kinetic traces of the PIA associated with 4CzIPN from 800 to 830 nm in 4CzIPN:CBP (black circles, Fig. 3e) and 4CzIPN:TTM-3PCz:CBP (red squares), we observed reductions in both the prompt and delayed lifetimes, from 12.1 to 7.8 ns and 2.5 μs to 1.0 μs, respectively, from the presence of TTM-3PCz. This provides further evidence for energy transfer from 4CzIPN $T_1$ (delayed

kinetic), and additionally from 4CzIPN $S_1$ (prompt kinetic), to form TTM-3PCz $D_1$.

Triplet–doublet energy transfer from 4CzIPN, a TADF molecule, can be attributed to a hyperfluorescent-type mechanism by breakout from $S_1$-$T_1$ ISC and rISC cycles[36],

$$D(T_1) + A(D_0) \rightarrow D(S_1) + A(D_0) \rightarrow D(S_0) + A(D_1) \quad (6)$$

i.e., 4CzIPN reverse intersystem crossing, then singlet–doublet Förster transfer, or triplet–doublet direct Dexter-type mechanism[37,38] as given in Eq. (4). Both mechanisms lead to reduced $T_1$ lifetime. In order to distinguish the energy transfer mechanisms, we have performed temperature dependence studies (50–293 K) on trPL of 4CzIPN:CBP (Supplementary Fig. 10) and 4CzIPN:TTM-3PCz:CBP (Supplementary Fig. 11). In both films there is negligible temperature dependence on trPL up to 100 ns, which we define as the prompt emission; we classify light emission from 100 ns onwards as delayed-type. The ratio of integrated delayed emission at different temperatures ($T$) with respect to the integrated value at 293 K is shown in Fig. 3f (i.e., delayed PL($T$)/delayed PL($T$ = 293 K)). The delayed PL ratio is reduced in 4CzIPN:CBP films compared to 4CzIPN:TTM-3PCz:CBP, falling to 0.2 and 0.8 at 50 K, respectively. This supports a Dexter-type triplet–doublet energy transfer channel in 4CzIPN:TTM-3PCz:CBP, with lower activation energy than reverse intersystem in 4CzIPN:CBP for thermally activated delayed fluorescence. However, the signal:noise for delayed PL

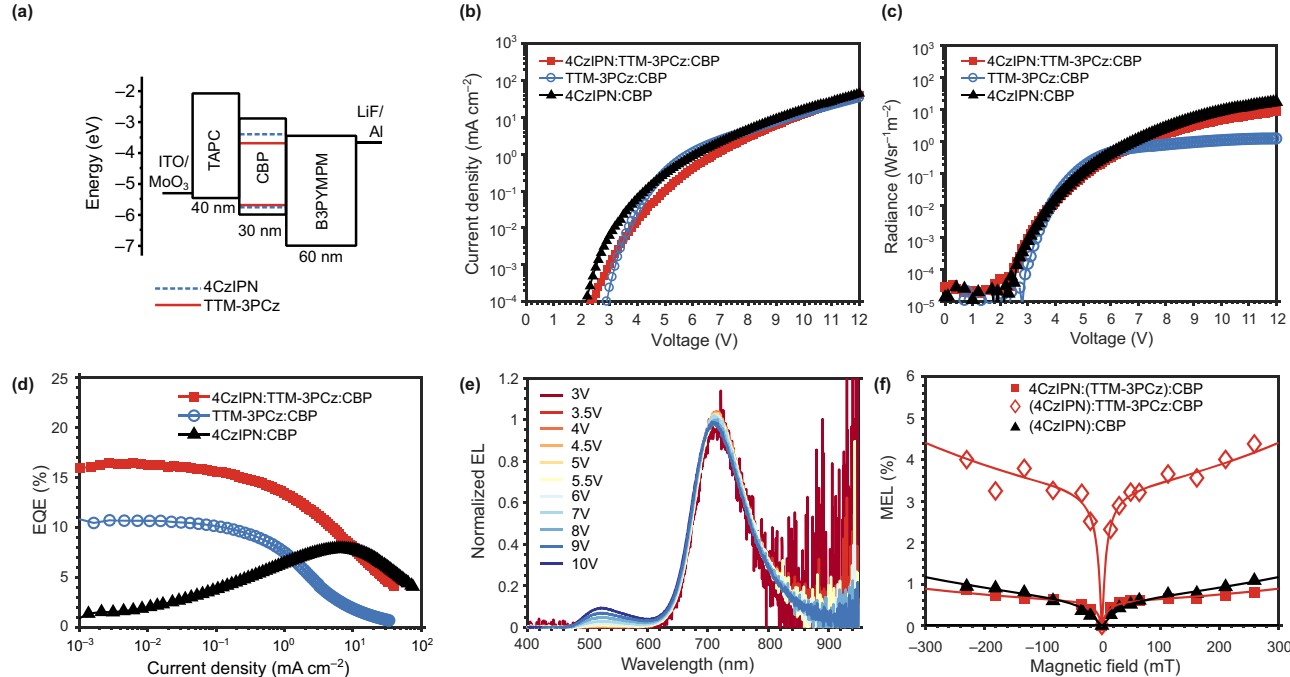

**Fig. 4 4CzIPN and TTM-3PCz organic light-emitting diodes. a** Device architecture for OLEDs with varying emissive layer: 4CzIPN:TTM-3PCz:CBP, 4CzIPN:CBP; TTM-3PCz:CBP. **b–d** Current density–voltage (*J–V*), radiance–voltage, EQE–current density (from $10^{-3}$ mA/cm$^2$) curves for OLEDs. **e** Normalised EL profiles for 4CzIPN:TTM-3PCz:CBP OLEDs with varying voltage, and 4CzIPN and TTM-3PCz emission contributions. **f** Magneto-electroluminescence (MEL) studies of TTM-3PCz (red squares) and 4CzIPN (red diamonds) emission in 4CzIPN:TTM-3PCz:CBP OLEDs; 4CzIPN emission in 4CzIPN:CBP (black triangles). OLED devices were biased at 8 V. MEL studies show different magnetic field dependencies for 4CzIPN and TTM-3PCz emission from 4CzIPN:TTM-3PCz:CBP devices, which supports Dexter triplet–doublet energy transfer and not the hyperfluorescence mechanism of 4CzIPN triplet exciton energy harvesting.

ratio varies in 4CzIPN:TTM-3PCz:CBP with changing temperature, restricting further quantitative analysis.

From the film photophysical studies, we have demonstrated efficient singlet–doublet and triplet–doublet energy transfer in 4CzIPN:TTM-3PCz:CBP from picosecond to microsecond timescales, which we have attributed to Förster and Dexter mechanisms that enable luminescent TADF:radical films with emission from radical $D_1$.

**Radical OLEDs and magneto-electroluminescence studies.** Following our demonstration of singlet–triplet–doublet energy transfer photophysics, we aimed to exploit these processes in more efficient radical-based OLED designs. We fabricated TADF:radical OLEDs using the device structure in Fig. 4a. B3PYMPM (4,6-bis(3,5-di(pyridine-3-yl)phenyl)-2-methylpyrimidine) and TAPC (1,1-bis[[(di-4-tolylamino)phenyl]cyclohexane) were used as electron transport and hole transport layers, respectively. The emissive layer (EML) was 4CzIPN:TTM-3PCz:CBP (0.25:0.03:0.72)—the same composition as the photophysics studies. Single-dopant OLEDs were also fabricated where EML was 4CzIPN:CBP (0.25:0.75) for TADF reference devices; and EML was TTM-3PCz:CBP (0.03:0.97) for radical reference OLEDs.

The current density–voltage (*J–V*), radiance–voltage and EQE plots for the 4CzIPN:TTM-3PCz:CBP (red squares), 4CzIPN:CBP (black triangles) and TTM-3PCz:CBP (blue circles) OLEDs are shown in Fig. 4b–d. We found that the turn-on voltages decrease from 2.9 V (TTM-3PCz:CBP device) to 2.3 V (4CzIPN:TTM-3PCz:CBP) to 2.2 V (4CzIPN:CBP). Here, we define the turn-on voltage to be that corresponding to current density >0.1 μA/cm$^2$, above the electrical noise level of the devices. The trend in turn-on voltage suggests that the inclusion of the TADF sensitiser leverages more energy-efficient doublet exciton formation in

electroluminescence. However, the higher turn-on voltage for TADF:radical OLEDs compared to TADF, and different *J–V* profiles in Fig. 3b, imply that both CBP and 4CzIPN mediate some electrical excitation of TTM-3PCz in TADF:radical devices. If all doublet electroluminescence originated by energy transfer from TADF sensitisation as in Fig. 1d, the *J–V* curves and turn-on voltages would be identical for 4CzIPN:CBP and 4CzIPN:TTM-3PCz:CBP OLEDs.

We note there is a plateau in maximum radiance of ~1 W sr$^{-1}$ m$^{-2}$ from 5 V for TTM-3PCz:CBP devices in Fig. 4c; radiance values up to 10 W sr$^{-1}$ m$^{-2}$ are achievable in 4CzIPN:TTM-3PCz:CBP. At voltages higher than 5 V, there is an increasing component of 4CzIPN emission in the total EL of 4CzIPN:TTM-3PCz:CBP OLEDs. At 10 V the EL from the device contains 89% TTM-3PCz and 11% 4CzIPN contributions. The higher radiance at 10 V for 4CzIPN:TTM-3PCz:CBP (5.0 W sr$^{-1}$ cm$^{-2}$) compared to TTM-3PCz:CBP (1.1 W sr$^{-1}$ cm$^{-2}$) in Fig. 4c is therefore consistent with increasing energy transfer contribution from electrically excited 4CzIPN. The EL profile at 10 V in Fig. 4e resembles the steady-state PL profile for 4CzIPN:TTM-3PCz blends (Supplementary Fig. 8).

Figure 4d shows that there is substantial increase in maximum EQE on going from 4CzIPN:CBP (7.8%) and TTM-3PCz:CBP (10.7%) devices to 4CzIPN:TTM-3PCz:CBP (16.4%) OLEDs. The EQE is evaluated for the total EL output. We note that the 25% wt. 4CzIPN:CBP reference device shown here has lower EQE than previous reports with 3% wt. 4CzIPN concentration due to exciton self-quenching effects[8,33]. The high 4CzIPN concentration is necessary to promote charge trapping at the TADF component in 4CzIPN:TTM-3PCz:CBP blends. Here the higher EQE on going from 4CzIPN:CBP to 4CzIPN:TTM-3PCz:CBP OLEDs suggests efficient energy transfer from 4CzIPN to TTM-3PCz, leading

to performance that is not limited by the EL efficiency of the 4CzIPN:CBP device. $J_0$, the critical current density that corresponds to the device current at half the maximum EQE, increases from 2.1 mA cm$^{-2}$ for TTM-3PCz:CBP to 9.5 mA cm$^{-2}$ for 4CzIPN:TTM-3PCz:CBP. The better roll-off and sustained EL efficiency in 4CzIPN:TTM-3PCz:CBP OLEDs is also attributed to an increasing contribution of 4CzIPN energy transfer to the EL at higher current densities. At lower voltages (<5 V) and current densities (<0.1 mA cm$^{-2}$), the EL shows TTM-3PCz emission only (Fig. 4e). We performed studies to obtain the device's half-lifetime, T50 (time for luminance to fall to half of the initial value under a constant current density). The T50 of energy transfer-type 4CzIPN:TTM-3PCz:CBP OLEDs was found to be 42 min at 0.4 mA/cm$^2$ (see Supplementary Fig. 7), indicating some improvement over charge-trapping-type devices that we have previously reported for radical OLEDs with TTM-derivative:host EML (10 min at 0.1 mA/cm$^2$)[25].

Magneto-electroluminescence (MEL) and magnetoconductance (MC) studies have been performed on the 4CzIPN:CBP and 4CzIPN:TTM-3PCz:CBP devices. The devices were biased at 8 V and the data for magneto-EL and magnetoconductance were collected simultaneously. In 4CzIPN:CBP devices, MEL and MC profiles show enhanced EL and current density upon application of magnetic field (Fig. 4f and Supplementary Fig. 9). The profiles are fitted to double Lorentzian functions that capture low (<10 mT) and high (>10 mT) magnetic field effects (MFEs). The low field dependence is characteristic of magnetic field effects on hyperfine-mediate spin mixing of singlet and triplet polaron pair[39], the precursors of excitons, which affect the ratio of singlet and triplet exciton formation. High field effects can arise from triplet exciton–polaron quenching and singlet–triplet dephasing effects[40,41]. MFEs of 4CzIPN:CBP devices are positive and show typical behaviour for MEL and MC from non-radical dopant systems, as previously reported[42].

In TADF:radical OLEDs (4CzIPN:TTM-3PCz:CBP) we have studied magnetic field effects on EL from TTM-3PCz (680–800 nm) and 4CzIPN (500–550 nm) emission contributions. We observe positive magnetic field effects for both TTM-3PCz and 4CzIPN contributions, which indicates that the main magnetic field sensitivity originates from hyperfine-mediated spin mixing of singlet–triplet polaron pairs, as found in the TADF-only devices. However the size of MEL for 4CzIPN (+4% at 250 mT) and TTM-3PCz (+1% at 250 mT) emission components are different in TADF:radical OLEDs. We consider that non-identical MEL profiles for 4CzIPN and TTM-3PCz emission in 4CzIPN:TTM-3PCz:CBP devices supports a Dexter triplet–doublet energy transfer mechanism because an identical field sensitivity would be expected for the 4CzIPN and TTM-3PCz MEL in TADF:radical hyperfluorescent-type devices.

We have demonstrated efficient energy transfer of 4CzIPN singlet and triplet excitons to obtain emissive doublet excitons of TTM-3PCz. In trPL studies we observed more rapid light emission in 4CzIPN:TTM-3PCz:CBP blends than 4CzIPN:CBP, as up to 95% and 50% of photons are emitted by 1 μs, respectively. TA measurements revealed singlet–doublet and triplet–doublet energy transfer on 10–100 ns and 100 ns–1 μs timescales, though the observed timescale of triplet transfer is limited by the time taken for intersystem crossing to take place on 4CzIPN and, as a spin-allowed process, may be faster than this. OLEDs with 4CzIPN:TTM-3PCz:CBP emissive layer were demonstrated with max EQE = 16.4% and $J_0$ = 9.5 mA/cm$^2$, which outperforms TTM-3PCz:CBP (max EQE = 10.7%, $J_0$ = 2.1 mA/cm$^2$) for the same charge transport layer architecture. With also an order of magnitude improvement in device stability, the energy transfer-type radical OLEDs therefore show a substantial improvement in device characteristics compared to

previous reports of charge-trapping radical OLEDs. The MEL results allow us to rule out a fully hyperfluorescence-type (Eq. (6)) mechanism for EL, and support Dexter-type $T_1$-$D_1$ energy transfer pathways enabled by organic radicals, here TTM-3PCz. We highlight that Dexter triplet–triplet transfer from energy donor to acceptor is a loss route for light emission with non-radicals, and must be suppressed in energy transfer devices using non-radical fluorescent emitters, for example, hyperfluorescence-type devices[15]. However fluorescent radical (doublet) emitters can exploit the triplet–doublet energy transfer pathway for radical OLEDs as we have demonstrated here, without a lower-lying radical 'triplet state' that must be avoided for emission losses. In future work, our device concepts can be used in improved material combinations for more efficient energy transfer with reduced exciton quenching, and with increased radical luminescence for advancing the performance beyond this starting point. By unlocking new energy transfer channels, an optoelectronic design for improved radical-based light-emitting devices is enabled by their unpaired electron spin properties.

## Methods

**Materials.** TTM-3PCz precursor was synthesised by Suzuki coupling of tris(2,4,6-trichlorophenyl)methane (HTTM) and 4,4,5,5-tetramethyl-1,3,2-dioxaborolan-2-yl-3PCz[20]. In this procedure, TTM-3PCz radicals were generated from the precursor by treatment with potassium *t*-butoxide in tetrahydrofuran, followed by oxidation with *p*-chloranil. 4CzIPN, TAPC, B3PYMPM, CBP of sublimed grade and other OLED materials were obtained from Ossila, Xi'an Polymer Light and Lumtec.

**Photophysics.** TrPL and TA studies were performed on home-built setups powered by a Ti:sapphire amplifier (Spectra Physics Solstice Ace, 100 fs pulses at 800 nm, 7 W output at 1 kHz). TrPL profiles were recorded using an Andor spectrometer setup with electrically gated intensified CCD camera (Andor SR303i; Andor iStar). Sample excitation with 400 nm pump pulse was provided by frequency-doubled 800 nm pulse from Ti:sapphire amplifier in trPL and short-time (ps–ns) TA studies. Short-time TA studies with 600 nm excitation were achieved from the wavelength tuneable output of TOPAS optical parametric amplifier (Light Conversion), which was pumped by the 800 nm laser pulses from the Ti:sapphire amplifier. Long-time (ns–μs) TA studies were performed with 355 nm pump pulses from an Innolas Picolo 25. Probe pulses for TA were obtained from non-collinear optical parametric amplifier (NOPA) systems for the visible (500–780 nm), near-infrared (830–1000 nm) and infrared (1250–1650 nm) wavelength ranges. The NOPA probe pulses were divided into two identical beams by a 50/50 beamsplitter; this allowed for the use of a second reference beam for improved signal:noise. The probe pulse for the UV (350–500 nm) region was provided by a white light supercontinuum generated in a CaF₂ crystal. The probe pulses were detected by Si (Hamamatsu S8381-1024Q) and InGaAs (Hamamatsu G11608-512DA) dual-line array with a custom-built board from Stresing Entwicklungsbüro.

**Device fabrication and characterisation.** Organic semiconductor films and devices were fabricated by vacuum-deposition processing (<6 × 10⁻⁷ torr) using an Angstrom Engineering EvoVac 700 system. Current density, voltage and electroluminescence characteristics were measured using a Keithley 2400 sourcemeter, Keithley 2000 multimeter and calibrated silicon photodiode. The EL spectra were recorded by an Ocean Optics Flame spectrometer. Magneto-EL measurements were performed with Andor spectrometer (Shamrock 303i and iDus camera) for modulation of EL in presence of magnetic field applied by GMW 3470 electromagnet.

**Reporting summary.** Further information on research design is available in the Nature Research Reporting Summary linked to this article.

## Data availability

The data generated in this study have been deposited in the figshare database under the accession code: https://doi.org/10.6084/m9.figshare.17026607.v1.

## Code availability

Code used to analyse data in this manuscript are available from the corresponding author upon reasonable request.

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

## Acknowledgements
J.D., Z.C. and F.L. are grateful for financial support from the National Natural Science Foundation of China (grant no. 51925303). E.W.E. is grateful to the Leverhulme Trust for an Early Career Fellowship; and the Royal Society for a University Research Fellowship (grant no. URF\R1\201300). TJHH thanks the Royal Society for a University Research Fellowship (grant no. URF\R1\201502). WKM and the Centre for Advanced Electron Spin Resonance is supported by EPSRC (EP/L011972/1). F.L. is an academic visitor at the Cavendish Laboratory, Cambridge, and is supported by the Talents Cultivation Pro-gramme (Jilin University, China). A.J.G. and RHF acknowledge support from the Simons Foundation (grant no. 601946) and the EPSRC (EP/M01083X/1 and EP/M005143/1). This project has received funding from the ERC under the European Union's Horizon 2020 research and innovation programme (grant agreement no. 670405 and 101020167).

## Author contributions
E.W.E. and F.L. fabricated thin films and OLED devices, which were characterised by photoluminescence, *J–V*-radiance measurements and magnetic field studies. AJG performed transient absorption measurements. A.J.G. and E.W.E. carried out the transient PL measurements. Q.G. conducted OLED time dependence studies. J.D. and Z.C. synthesised the radical materials. TJHH formulated theory on the photophysical mechanisms. W.K.M. conducted spin physics studies. EWE, RHF and FL conceived the project and supervised the work. The results were analysed and the manuscript was written with input from all authors.

## Competing interests
The authors declare no competing interests.
