## [Peer Review File · Nature Communications]

Peer review comments, first round –

Reviewer #1 (Remarks to the Author):

Authors report radical (doublet)-based organic light-emitting diodes (OLEDs) using a double-dopant emissive layer comprising a TADF donor (4CzIPN) and a TTM-3PCz radical acceptor. It was demonstrated that fast singlet-doublet and triplet-doublet energy transfer from donor to acceptor takes place and thereby realized highly efficient and stable radical OLEDs that are superior to both the TADF and radical-only devices. All experimental and mechanistic studies were carefully done and supported the underlying principle. This study following previous works by the authors again represents a potential of radical OLEDs to improve device stability and a new design concept of OLED devices. Therefore, this manuscript can be accepted for publication in this journal.

A minor query: the EQE of 4CzIPN:CBP device (in Figure 4) appears to be relatively low compared to those of reported devices elsewhere. Is there any reason for this?

Reviewer #2 (Remarks to the Author):

The authors report the use of a doublet emitter in combination with a TADF-molecule in an OLED structure that shows more efficient emission than either component alone. This is attributed to singlet and triplet energy transfer from the TADF-molecule to the doublet emitter molecule. While the overall performance of the OLED is still moderate (15% EQE at 700nm with 45 min lifetime), this is a major conceptual advance in the search for new avenues to circumvent the limitations imposed by spin statistics. The approach itself is original, novel, and indicates a route for others to follow up and advance further. The evidence put forward to argue the singlet and triplet energy transfer is convincing. The paper is also very well written and accessible to those new in this particular topic. I recommend publication of the article. I note below a few minor points regarding the presentation of the paper.

- Page 4, first paragraph: The notation is first used where D denotes a doublet, whereas up to then D (bold) denoted the donor. It would help the reader if you point this out, so that he does not need to puzzle out what D0 and D1 is as opposed to D(T1).

- Figure 1c,d: To me, the bulbs were confusing, in particular since the energy transfer itself is NOT luminescent. I would recommend omitting them.

Reviewer #3 (Remarks to the Author):

The manuscript by Friend, Li, and co-workers presents an alternative concept towards the OLED efficiency enhancement. Unlike the conventional approaches, exploiting the wealth of doped and non-doped TADF materials, as well as the less common hyperfluorescence approach, employing the Forster transfer, the authors report on the novel concept of triplet to doublet transition, utilizing the doublet states of the radical compound TTM-3PCz. Dexter mechanism, which is generally avoided as a parasitic loss pathway in the case of conventional OLEDs, is exploited to support the T1-D1 energy transfer route enabled by organic radicals. Not only do the authors confirm a superior performance of a TADF: radical emitter:host-based OLED when compared to the conventional TAF:host system and radical emitter:host, but also shorter TADF lifetime is achieved, hence enhancing the OLED stability. Furthermore, the results are well backed up by the time-resolved photoluminescence, magneto-conductance, and magneto-electroluminescence measurements. Additionally, fundamental photophysical processes in radical emitters, TADF, Dexter, and Forster transfers are revisited and explained, being in line with the reported results. I recommend publication of this manuscript after a minor revision and clarification of the following:

1. The authors have chosen 4CzIPN as a TADF emitter and TTM-3PCz as a radical emitter. Which are the prospects for other TADF and radical emitters? Are there any design rules?
2. The authors report on the EQE of 16.4%. Is there any way to improve this value?
3. What is the authors' view on the commercial suitability of TADF:radical emitter systems as

reported here?

Reply to Reviewers:

'Singlet and triplet to doublet energy transfer: improving organic light-emitting diodes with radicals'

We thank the reviewers for their comments on our manuscript. In our reply below, reviewers' comments are indicated in blue, our response in black, and revised text for manuscript and supplementary information in red.

Reviewer 1

We are pleased that the Reviewer found: 'All experimental and mechanistic studies were carefully done and supported the underlying principle. This study following previous work by the authors again represents a potential of radical OLEDs to improve device stability and a new design concept of OLED devices. Therefore, this manuscript can be accepted for publication in this journal'. We address Reviewer 1 comments as follows:

Comment 1.1: 'A minor query: the EQE of 4CzIPN:CBP device (in Figure 4) appears to be relatively low compared to those of reported devices elsewhere. Is there any reason for this?'

The EQE of 4CzIPN:CBP device is lower than the benchmark results published in Uoyama et al, Nature, 492, 234-238 (2012), because in our work 4CzIPN is doped at the concentration of 25% rather than 3% reported by Uoyama et al. and other groups elsewhere. The higher doping concentration leads to exciton self-quenching and reduced luminescence efficiency in 4CzIPN:CBP devices. It is necessary to dope 4CzIPN at this high concentration to promote charge trapping at TADF sites when combined with radicals in 4CzIPN:TTM-3PCz:CBP, so that energy transfer occurs to radicals and we obtain doublet emission from our devices. In the manuscript we describe this on Page 7-8: 'The composition is based on the starting point of our previous work on TTM-3PCz OLEDs²⁰, which here allows us to test energy transfer mechanisms in proof-of-principle studies. 4CzIPN and TTM-3PCz were blended in CBP (4,4'-bis(N-carbazolyl)-1,1'-biphenyl) to reduce the effects of exciton self-quenching³³, and with higher doping of 4CzIPN than the radical to promote charge trapping at the TADF sites and subsequent energy transfer to TTM-3PCz for light emission.').

To clarify the relevant ideas, the following text is added on Page 13: 'We note that the 25% wt. 4CzIPN:CBP reference device shown here has lower EQE than previous reports with 3% wt. 4CzIPN concentration due to exciton self-quenching effects.^{8,33} The high 4CzIPN concentration is necessary to promote charge trapping at the TADF component in 4CzIPN:TTM-3PCz:CBP blends. Here the higher EQE on going from 4CzIPN:CBP to 4CzIPN:TTM-3PCz:CBP OLEDs suggests efficient energy transfer from 4CzIPN to TTM-3PCz, leading to performance that is not limited by the EL efficiency of the 4CzIPN:CBP device.'

Reviewer 2

We are pleased that the Reviewer found: 'this is a major conceptual advance in the search for new avenues to circumvent the limitations imposed by spin statistics. The approach itself is original, novel, and indicates a route for others to follow up and advance further. We address Reviewer 1 comments as follows:

Comment 2.1: 'Page 4, first paragraph: the notation is first used where D denotes a doublet, whereas up to then D (bold) denoted the donor. It would help the reader if you point this out, so that he does not need to puzzle out what D₀ and D₁ is as opposed to D(T₁)'.

We have included text to help the reader distinguish the doublet electronic states D₀, D₁ and energy donor notation **D**(T₁).

'(Note that D_x denotes doublet electronic states, and bold **D** denotes energy donor).' on page 4, paragraph 2.

Comment 2.2: 'Figure 1c,d: To me, the bulbs were confusing, in particular since the energy transfer itself is NOT luminescent. I would recommend omitting them'.

We have removed the cartoon bulbs in Figure 1 to avoid confusion on page 6.

Reviewer 3

We are pleased that the Reviewer found 'a superior (OLED) performance' in our work and that 'the results are well backed up' to 'recommend publication of this manuscript after a minor revision and clarification of the following:'

Comment 3.1: '1. The authors have chosen 4CzIPN as a TADF emitter and TTM-3PCz as a radical emitter. What are the prospects for the other TADF and radical emitters? Are there any design rules?'

We have chosen 4CzIPN and TTM-3PCz as benchmark TADF and radical emitters to demonstrate this new device concept using energy transfer. As reported by Reviewer 2, our work 'indicates a route for others to follow up and advance further.'

We consider there are general design rules, and now explicitly state these as conditions 1. and 2. on Page 5 to make this clearer and benefit the reader.

'General design rules are formulated: singlet (S₁) and triplet (T₁) excitons of **D** can transfer energy to the doublet (D₁) of **A** for efficient doublet emission where

1. The singlet and triplet energy levels of the donor are higher than the D₁ state of the acceptor, i.e. $E(\mathbf{D}, S_1) > E(\mathbf{A}, D_1)$ and $E(\mathbf{D}, T_1) > E(\mathbf{A}, D_1)$ where $E(\mathbf{D}, S_1)$ and $E(\mathbf{D}, T_1)$ are the S₁ and T₁ exciton energies of **D**, and $E(\mathbf{A}, D_1)$ is the radical **A** D₁ exciton energy;

2. The donor-cation/acceptor-anion, $D^{*+}A^{*-}$ or donor-anion/acceptor-cation, $D^{*-}A^{*+}$ states must be higher energy than the radical D_1 -exciton, i.e. $E(D^{*+}A^{*-}) > E(A, D_1)$ and $E(D^{*-}A^{*+}) > E(A, D_1)$.'

Comment 3.2: '2. The authors report on the EQE of 16.4%. Is there any way to improve this value?'

The EQE of the device can be improved with more efficient energy transfer process and more efficient radical emission (i.e. photoluminescence quantum yield, PLQY). We are exploring other energy donor systems as future work for publication, using materials other than 4CzIPN that do not undergo exciton quenching at high concentration and even allowing the materials to act as neat hosts of radicals for more efficient energy transfer. For more efficient radical emitters, there are ongoing efforts to develop new highly luminescent doublet materials, which could also be used to increase the EQE of devices based on the radical energy transfer concept. We now highlight this opportunity in the Conclusion on page 15:

'In future work our device concepts can be used in improved material combinations for more efficient energy transfer with reduced exciton quenching, and with increased radical luminescence for advancing the performance beyond this starting point.'

Comment 3.3: '3. What is the authors' view on the commercial suitability of TADF:radical emitter systems as reported here?'

The commercial opportunity for the TADF:radical devices are currently in red or infrared devices, using TTM-based radical emitters where the TTM itself has 570 nm emission and all derivatives have redder emission. We highlight a new device concept that could be used to advance the general performance of radical devices, as discussed in the revised text in reply to Comment 3.2. We do consider this has real promise, but of course this will require further engineering development work to fully meet commercial standards.

We hope you find these revisions satisfactory for publication. Please do not hesitate to contact us if you need further information.

Yours sincerely,
Richard Friend
Feng Li

Peer review comments, second round –

Reviewer #1 (Remarks to the Author):

The revision of the original manuscript was properly done so it can be accepted for publication in this journal.

Reviewer #3 (Remarks to the Author):

The concerns I raised were addressed both in the manuscript and in the letter.
I recommend the publication of this article.